# Epigenetics of *SFRP1*: The Dual Roles in Human Cancers

**DOI:** 10.3390/cancers12020445

**Published:** 2020-02-14

**Authors:** Rashidah Baharudin, Francis Yew Fu Tieng, Learn-Han Lee, Nurul Syakima Ab Mutalib

**Affiliations:** 1UKM Medical Molecular Biology Institute (UMBI), Universiti Kebangsaan Malaysia, Cheras, Kuala Lumpur 56000, Malaysia; ieda_baharudin@yahoo.com (R.B.); francistieng@yahoo.com.my (F.Y.F.T.); 2Novel Bacteria and Drug Discovery Research Group, Microbiome and Bioresource Research Strength, Jeffrey Cheah School of Medicine and Health Sciences, Monash University Malaysia, Subang Jaya 47500, Malaysia

**Keywords:** *SFRP1*, DNA Methylation, cancer biomarkers, Wnt signaling pathway, gene expression, microRNA

## Abstract

Secreted frizzled-related protein 1 (SFRP1) is a gene that belongs to the secreted glycoprotein SFRP family. *SFRP1* has been classified as a tumor suppressor gene due to the loss of expression in various human cancers, which is mainly attributed by epigenetic inactivation via DNA methylation or transcriptional silencing by microRNAs. Epigenetic silencing of *SFRP1* may cause dysregulation of cell proliferation, migration, and invasion, which lead to cancer cells formation, disease progression, poor prognosis, and treatment resistance. Hence, restoration of *SFRP1* expression via demethylating drugs or over-expression experiments opens the possibility for new cancer therapy approach. While the role of *SFRP1* as a tumor suppressor gene is well-established, some studies also reported the possible oncogenic properties of *SFRP1* in cancers. In this review, we discussed in great detail the dual roles of *SFRP1* in cancers—as tumor suppressor and tumor promoter. The epigenetic regulation of *SFRP1* expression will also be underscored with additional emphasis on the potentials of *SFRP1* in modulating responses toward chemotherapeutic and epigenetic-modifying drugs, which may encourage the development of novel drugs for cancer treatment. We also present findings from clinical trials and patents involving *SFRP1* to illustrate its clinical utility, extensiveness of each research area, and progression toward commercialization. Lastly, this review provides directions for future research to advance *SFRP1* as a promising cancer biomarker.

## 1. Introduction

Secreted frizzled-related protein 1 (*SFRP1*) is a gene that belongs to the secreted glycoprotein SFRP family. This family is also composed of another four secreted glycoproteins, namely, SFRP2, SFRP3, SFRP4, and SFRP5, which have been identified in humans [1]. Among the five members of the SFRP family, *SFRP1* has been extensively studied in human cancers. This gene is located within the 8p11.21 chromosome region [2] and encodes a secreted protein with 314 amino acids (35.4 kDa) [3]. The SFRP1 protein harbors two independent structural domains, namely the carboxy-terminal netrin (NTR) domain and an amino-terminal cysteine-rich domain (CRD). CRD domain is homologous to the putative Wnt-binding site of frizzled (Fz) receptors because it contains ten cysteines with a pattern of five disulfide bridges that is similar to the CRD of Fz [4]. Therefore, SFRP1 can act as a modulator of the Wnt signaling pathway.

*SFRP1* has been classified as a tumor suppressor gene due to the loss of its expression in many human cancers. This may cause dysregulation of cell proliferation, migration, and invasion, which eventually lead to cancer cells’ formation. The loss of *SFRP1* expression is associated with the early development of colorectal cancer (CRC) as well as prostate cancer, and is linked with disease recurrence in renal cell cancer [5].

Various mechanisms have been implicated in the loss of *SFRP1* including epigenetic and genetic regulation. Endogenous *SFRP1* expression increases in a dose-dependent manner after demethylating treatment, signifying DNA methylation as the main mechanism that is responsible for the silencing of *SFRP1* [6]. Therefore, targeting DNA methyltransferase activity represents a promising strategy to reduce or reverse the methylation in the *SFRP1* promoters. Previously, HDAC inhibitor (HDACi); romidepsin and DNA methyltransferase inhibitor (DNMTi); and 2′-deoxy-5-azacytidine (Decitabine) were used to restore *SFRP1* expression in cancer cells (please refer to Section 4 below). The restoration of *SFRP1* sensitized the cisplatin-resistant laryngeal carcinoma cells [7]. Moreover, reversing *SFRP1* methylation using Decitabine suppressed cell proliferation, invasion, and migration of nasopharyngeal cancer [8]. Taken together, these strategies highlight the potential of using epigenetic drugs for cancer treatment. While the role of *SFRP1* as a tumor suppressor gene is well-established, some studies also reported the possible oncogenic properties of *SFRP1* in cancers. *SFRP1* is highly expressed in the basal-like subtype [9] as well as in the triple-negative breast cancer (TNBC) [10]. Similarly, *SFRP1* was also found to be over-expressed in metastatic renal cell carcinomas but not in primary tumors [11], and this was further verified in gastric cancer cells [12,13].

A comprehensive and general review on the *SFRP* family was published more than five years ago [14]. While *SFRP2* and *SFRP4* were more recently reviewed [15,16], an updated review that recapitulates the association between *SFRP1* and chemoresistance is limited. A pan-cancer analysis suggested that *SFRPs* are strongly correlated with patient survival, but there is an inconsistency between family members and cancer types [17]. A systematic review and meta-analysis of the *SFRP* family also revealed that *SFRP1* hypermethylation was significantly associated with cancer risk [18]. Therefore, in this review, epigenetic regulation of *SFRP1* expression will be highlighted with additional emphasis on the potentials of *SFRP1* in modulating responses toward chemotherapeutic and epigenetic-modifying drugs. We also provide the latest evidence of the divergent roles of *SFRP1* in tumorigenesis that may encourage the development of novel drugs for cancer treatment by targeting *SFRP1*.

## 2. Epigenetic Inactivation and Genomic Alterations in *SFRP1* Gene

The expression of *SFRP1* mRNA is detectable in all tested human tissues and broad expression was observed in the endometrium [19], ovary [20], colon [21], prostate [22], and breast [17]. The Cancer Genome Atlas (TCGA) revealed reduced expression of *SFRP1* various cancers including breast, colorectal, lung, bladder urothelial carcinoma, cervical squamous cell carcinoma, head and neck squamous cell carcinoma, glioblastoma multiforme, kidney renal clear cell carcinoma, stomach adenocarcinoma, and endometrium cancer compared to the normal tissues [23]. The downregulation of *SFRP1* expression in cancer can be regulated through different mechanisms such as non-coding RNA (ncRNA), DNA methylation, allelic imbalance, or genomic alterations.

### 2.1. MicroRNA-27a (miR-27a)

In general, ncRNA functions to regulate gene expression at the transcriptional and post-transcriptional level. Dysregulation of ncRNAs plays a significant role in tumor initiation and progression. MicroRNAs (miRNAs) are the widely studied type of ncRNA and are closely associated with cancer. Numerous miRNAs have been demonstrated to accelerate tumorigenesis by targeting *SFRP1*.

*SFRP1* is a prior target of miR-27a in several tumors [24,25]. miR-27a is upregulated in the gastric cancer cell line MGC803 [24], breast cancer cell lines BT-20, MCF7, T-47D, and MDA-MB-231 [26], as well as in osteosarcoma cell lines HOS, SaOS2, 143B, and MG63 [27]. This miRNA negatively regulated *SFRP1* by hindering its expression. In contrast, knockdown of miR-27a in gastric cancer cell line resulted in increased *SFRP1* mRNA and at the same time decreased the expression of Wnt and β-catenin. This indicates that miR-27a targets *SFRP1* to activate the Wnt/β-catenin signaling pathway. The activation of this signaling pathway remarkably increased the proliferation, migration, and invasion of gastric cancer cells [24].

Moreover, a higher level of miR-27a and low level of *SFRP1* were found in CRC tissues as compared to normal tissues. The authors verified the negative correlation of miR-27a with *SFRP1* in colon cancer by transfecting HCT-116 cells with miR-27a mimics [25]. The low expression of *SFRP1* resulted from the degradation of the mRNA through the binding of miR-27a to the *SFRP1* 3′-UTR and not likely from the inhibition of protein translation [25]. Mu et al. also proved that miR-27a could promote the proliferation and invasion of human osteosarcoma cells through *SFRP1*-dependent Wnt/β-catenin signaling pathway [28]. This study also demonstrated that knockdown of miR-27a induced the upregulation of *SFRP1* and suppressed the cell proliferation and invasion in osteosarcoma cell lines [28].

Only miR-27a was studied extensively in multiple cancers to date. There are other miRNAs targeting *SFRP1* being investigated to a lesser extent as displayed in Table 1.

### 2.2. DNA Methylation Regulates SFRP1 Expression

The *SFRP1* expression can be epigenetically regulated through promoter DNA methylation. Hypermethylation of the *SFRP1* promoter has been recognized as a common mechanism for downregulation of this gene in cancers [35,36,37]. Methylation inactivates the *SFRP1* expression by the addition of a methyl group (CH3) to the CG-rich region (CpG island), which is mainly located in the promoter regions of *SFRP1* [37]. The unmethylated CpG islands enhance the accessibility of *SFRP1* promoter to the transcription factor and other regulatory units such as enhancers. The presence of DNA methylation forms a heterochromatin structure and restricts the access of the transcription factor to bind to the promoter region [38]. This subsequently inhibits the initiation of gene transcription.

*SFRP1* hypermethylation and subsequently reduced mRNA expression are extensively studied in CRC. Meta-analysis using TCGA data showed that hypermethylation of *SFRP1* caused the downregulation of this gene and high *SFRP1* methylation is associated with poorer survival among CRC patients [36]. *SFRP1* methylation was also recently shown to be detectable in circulation [39], highlighting its potential as a non-invasive biomarker for CRC’s early detection. *SFRP1* gene methylation was significantly associated with its reduced mRNA expression [39]. Combined with *RUNX3* and carcinoembryonic antigen (CEA), this panel identified CRC with 89.41% sensitivity in tissue and 84.71% in serum [39].

In another study, the methylation of the *SFRP1* gene in bladder cancer tissues occurred more frequently than its adjacent normal tissues. They discovered that *SFRP1* methylation suppresses the gene expression and was associated with the pathogenesis of bladder cancer via the Wnt signaling pathway [37].

A study by Zhang et al. postulated that *SFRP1* acts as a potential biomarker for NSCLC because the epigenetic silencing of this gene was associated with lymph nodes metastasis and disease progression within a year after surgery [40]. Similarly, a meta-analysis in NSCLC discovered *SFRP1*’s transcriptional silencing as a consequence from promoter methylation, revealing a promising application of epigenetic therapy in NSCLC [41]. In addition, by analyzing TCGA data, Liu et al. [42] identified significant correlations between *SFRP1*_cg15839448 promoter methylation and *SFRP1* gene expression.

In gastric cancer, *SFRP1* methylation was associated with loss of *SFRP1* expression and occurred in the early event of this cancer. The epigenetic silencing of *SFRP1* was also significantly correlated with tumor stage and lymph node status [43]. To illustrate the methylation signatures of *SFRP1* among various cancers, the signatures from the TCGA datasets are displayed in Figure 1. Table 2 summarizes other published literature within six years pertaining to epigenetic regulation of *SFRP1* in human cancers.

### 2.3. Allelic Imbalance of SFRP1

Besides epigenetic mechanisms, genetic events may be involved in the deregulation of *SFRP1*. The *SFRP1* loci are commonly associated with loss of heterozygosity (LOH) in cancer, including hepatocellular carcinoma (HCC). A study by Huang and colleagues showed only 13% and 6.5% exhibit the allelic imbalance of the *SFRP1* loci D8S532 and D8SAC016868, respectively. They also discovered patients that exhibit allelic imbalance of *SFRP1* loci to experience a low expression of the gene [35]. This finding is supported by another study where positive LOH at the locus D8S532 is associated with downregulation of *SFRP1* expression [53]. However, the allelic imbalance of *SFRP1* loci in the HCC patients could not be a crucial event in suppressing *SFRP1* because it only involved low frequency of LOH. In the NSCLC, *SFRP1* expression is inactivated and 15 out of 40 patients exhibited allelic loss while 25 patients manifest LOH in the *SFRP1* locus [54]. Fewer studies are looking into the allelic imbalance of *SFRP1* since DNA methylation of the promoter *SFRP1* yield more insight into how *SFRP1* is downregulated.

### 2.4. Genomic Alterations in SFRP1

A query of 10,953 patients in 32 studies from TCGA Pan-Cancer reveals a varying degree of genomic alteration profiles (Figure 2) [55,56]. Among all of the alterations, amplification is the most frequent alteration observed, followed by deep deletion and mutation. *SFRP1* is only altered in 455 (4%) among the 10,953 patients, further supporting the notion that epigenetic mechanisms play a more important role. It has also been proven that in CRC, even though cancer-associated nonsense mutation that prematurely terminates protein translation at codon 151 (N150) was identified, point mutation is not the primary cause of *SFRP1* inactivation [57]. On the contrary, other evidence has shown that *SFRP1* mutations found in glioblastoma and CRC promote cancer by compromising the senescence-inducing activity of *SFRP1*, thus failing to antagonize Wnt signaling [58]. It is known that senescence is a two-edged sword: it can promote malignant transformation by modifying the cellular microenvironment or is a potential mechanism for a cell to avoid cancer development [59]. The role of *SFRP1* mutation in cancer-associated senescence warrants further research.

## 3. Role of *SFRP1* in Cancer Pathways

SFRP1 is implicated in various cancer-related pathways. Many studies have delineated the involvement of SFRP1 as a negative modulator of the Wnt signaling pathway. Additionally, SFRP1 is also involved in Hedgehog and TGF-β signaling pathways.

### 3.1. Involvement of SFRP1 in the Wnt Signaling Pathway

SFRP1 modulates the Wnt signaling pathway through different modes of activation. Interestingly, NTR and CRD of SFRP1 are necessary for optimal Wnt inhibition. SFRP1 can antagonize Wnt activity by directly binding to the ligand of the Wnt protein through its NTR domain and eventually inhibits the interaction of the Wnt ligand to the Fz receptor [60]. However, one study has demonstrated that SFRP1 could also regulate the Wnt signaling pathway by circumventing the interaction with the Wnt ligand. They discovered that the binding of SFRP1 to β-catenin may inhibit the interaction of β-catenin with T-cell factor (TCF) in the nucleus, further blocking Wnt signaling activation [61].

Alternatively, SFRP1 could prevent Wnt signal transduction by interacting with the Fz receptor through their corresponding CRD motif, thereby preventing Wnt ligand interaction with the receptor [62]. In the absence of Wnt ligands, β-catenin molecules are combined with the destruction complex, which consists of scaffold proteins: axin, adenomatous polyposis coli (APC), glycogen synthase kinase (GSK-3β), and casein kinase 1 (CK1). In this state, the β-catenin molecules are phosphorylated by GSK-3β and CK1. Phosphorylation-mediated ubiquitination of β-catenin by β-TrCP and induced proteasomal degradation resulted in a low level of β-catenin in the cytoplasm. Subsequently, the cytoplasmic β-catenin could not be translocated into the nucleus. In the nucleus, the absence of β-catenin initiates a repressive complex containing T-cell factor/lymphoid enhancing factor (TCF/LEF) and transducin-like enhancer protein such as Groucho to recruit histone deacetylase (HDACs) in order to repress the Wnt target genes [60,63], thus, further inhibiting the proliferation and invasion of the cells.

The decline of SFRP1 expression promotes the binding of Wnt ligand to the Fz receptor [61]. Upon binding, the lipoprotein receptor-related protein (LRP) is phosphorylated by CK1 and GSK-3β, which then recruits Dishevelled (Dvl) proteins to the plasma membranes where they polymerize and activate. The activated Dvl polymers inactivate the destruction complex and stabilize β-catenin by suppressing the phosphorylation process, thus, leading to the accumulation of β-catenin in the cytoplasm, and its further entry into the nucleus. Nuclear β-catenin forms an active compound with TCF/LEF proteins by replacing Groucho proteins, and thereby activating the transcription and expression of Wnt target genes [64]. Figure 3 illustrates the multiple mechanisms of SFRP1 involvement in the Wnt signaling pathway.

### 3.2. Involvement of SFRP1 in the Hedgehog Signaling Pathway

The accumulation of β-catenin in the nucleus is a major indicator of activated Wnt signaling. However, some tumor cells may possess abrogated nuclear β-catenin specifically in the activated Hedgehog (Hh) signaling tumor cells [65]. The Hh signaling pathway is important for cellular growth and differentiation during embryonic development [66]. Dysregulation of Hh signaling has been implicated in several cancers including esophageal [67], gastric [68], pancreatic [69], as well as liver [70]. Previous studies showed activated Hh signaling may attenuate Wnt activity through the high expression of the Wnt inhibitor, SFRP1 [63,71,72].

Hh pathway activation is achieved by the interaction of Hh ligands to the Patched receptor (Ptch), and thus allows the accumulation of the transducer smoothened (SMO) at the cell surface. The accumulated SMO stimulates downstream components of the signaling pathway including Gli1 molecules, which then translocate into the nucleus and further activate the Hh target genes [72,73]. *SFRP1* is a Hh target gene and is significantly regulated by *Gli1* [71]. This targeted gene comprises of a putative *Gli1* binding site, which allows the binding of *Gli1* to the promoter region of *SFRP1*. Thus, *SFRP1* expression is dependent on the *Gli1* transcript [65].

*SFRP1* is the hedgehog target that negatively regulates the Wnt signaling pathway. This has been confirmed by He et al. where knockdown of *SFRP1* in the Gli expressing cells is able to induce cytoplasmic β-catenin expression by Wnt-1 [71]. In addition, differentiated epithelial cells exhibit activated Hh and induce the *SFRP1* expression in the differentiated cells to hinder Wnt signaling activation within stem or progenitor cells [74].

### 3.3. Involvement of SFRP1 in TGF-β Signaling Pathway

SFRP1 over-expression restored the activity of GSK3β. Both SFRP1 and GSK3β are important to inhibit the Wnt signaling pathway [75]. However, emerging evidence showed the restoration of GSK3β in promoting the tumorigenesis is through other signaling pathways. It has been reported that GSK3β activates Rac family small GTPase-1 (Rac1) [76,77], and activation of this gene is involved in breast [78], colon [79], bladder [80], and gastric cancer [81], indicating a role of Rac1 in tumor development. Peng discovered SFRP1 over-expression activates GSK3β/Rac1 and simultaneously inhibits the pro-apoptotic effect of Smad3 in TGF-β signaling through the phosphorylation of the Smad3 linker region [82]. This study also demonstrated that the over-expression of SFRP1 activates TGF-β activity, which is correlated with cell proliferation, epithelial–mesenchymal transition (EMT), and invasion in gastric cancer cells [82].

On the contrary, SFRP1 was significantly downregulated in the TGF-β-induced EMT in lung cancer cell line A549. TGF-β suppresses SFRP1 expression and subsequently inactivates GSK3β by phosphorylating Serine 9 of GSK3β. Moreover, using an in vitro and in vivo model, ectopic expression of *SFRP1* was able to inhibit TGF-β activity through suppressing Wnt pathway [83]. In breast cancer, *SFRP1* knockdown activates the TGF-β signaling and further increases the expression of *ZEB2*, zinc finger clusters, that play a critical role in facilitating the EMT process [84]. Upon activation of TGF-β signaling, the downstream targets including Integrin β_3_ and PAI-1, which are responsible for cell migration and invasion, are also upregulated. Hence, this explains the migratory and invasive characteristics exhibited by *SFRP1* knockdown in breast cancer cells [85]. This study also suggested the knockdown of *SFRP1* can modulate TGF-β signaling not only through the Smad-dependent action but also over Smad-independent pathway through ERK1/2 phosphorylation [85].

### 3.4. Involvement of SFRP1 in Other Pathways

In androgen-dependent prostate cancer, loss of SFRP1 undergoes different pathways other than Wnt and Hh signaling to drive cancer cell proliferation. Kawano et al. discovered that SFRP1 represses androgen-receptor (AR) dependent transcription and subsequently inhibits cell proliferation in the androgen-dependent LNCaP cells [86]. SFRP1 negatively regulates AR through the CRD motif and binds with the Fz receptor to form SFRP1/Fz complexes. The inactivation of SFRP1 leads to the uncontrolled AR activation, which is involved in the pathogenesis of prostate cancer [87].

In addition, Bernemann and his colleagues also found that the underlying mechanism of SFRP1 in triple-negative breast cancer (TNBC) is independent of Wnt signaling pathway. They did not discover any changes in Wnt signaling activity and nuclear localization of β-catenin, besides they found the upregulation of genes that are involved in the migration process and downregulation of apoptotic genes after knockdown of *SFRP1* based on their gene ontology analysis result [10]. However, this finding is in contrast with Xu et al. whereby they found that the Wnt/β-catenin signaling pathway was enriched in the TNBC [88]. The contradiction may be due to the heterogeneity of the samples, which lead to the discovery of different pathways.

One study has suggested that SFRP1 plays an additional inhibitory role in breast cancer by blocking the activity of thrombospondin-1 (TSP1), which is involved in the modulation of adhesion and migration of cancer cells. This action is through the binding of NTR-related motif of SFRP1 to the N module of TSP1. They discovered the interaction between NTR of SFRP1 and TSP1 disrupts the adhesion and migration of MDA-MB-231 breast cancer cell via α3β1 integrin [89].

## 4. Clinical Utility of *SFRP1*

### 4.1. SFRP1 and Chemotherapy Response

As discussed earlier, Bernemann and colleagues found that knockdown of *SFRP1* is strongly correlated with TNBC subtypes. Moreover, they also found *SFRP1* could be used as a potential chemotherapeutic marker to stratify patients’ response toward chemotherapy. Knockdown of *SFRP1* rendered TNBC cell lines more resistance toward paclitaxel, cisplatin, and doxorubicin chemotherapy as well as radiotherapy [10].

HDAC inhibitor romidepsin and methyltransferase inhibitor decitabine are the FDA-approved epigenetic-modifying drugs for the treatment of myelodysplastic syndromes (MDS) and a subset of T cell lymphoma, respectively. In a study by Cooper and colleagues, epigenetic silencing of *SFRP1* was shown to contribute to renal and breast cancer cell survival [90]. Exposure of clear cell renal cell carcinoma (ccRCC) and TNBC cells to low doses of exogenous *SFRP1* resulted in dose-dependent inhibition of cancer cell through apoptosis induction [90]. Their findings also propose that *SFRP1* re-expression could be used as a biomarker for romidepsin/decitabine response.

Taxanes, such as docetaxel and taxol, are microtubule-stabilizing agents used as first-line agents in the treatment of advanced lung adenocarcinoma and other solid tumors since the 1990s [91]. However, poor response to treatment remains a challenge, and a new target for the treatment of taxane-resistant patients is urgently needed. Via microarray analysis, epigenetic inactivation of *SFRP1* by DNA methylation was implicated in taxane chemoresistance [92]. The authors further demonstrated that treatment with demethylating agent 5-azacytidine enhanced the sensitivity of lung cancer cell lines to taxanes, suggesting *SFRP1* methylation as a clinically relevant determinant of taxanes resistance in lung cancer patients [92]. Another study revealed that downregulation of *SFRP1* in laryngeal carcinoma is mediated by DNA methylation [7]. Subsequent treatment with a demethylating agent significantly increased the expression of this gene and enhanced the sensitivity of laryngeal carcinoma cells to cisplatin through inhibition of *NHE1* gene.

With regards to hematological malignancy, epigenetic silencing of *SFRP1* is infrequently observed in chronic myeloid leukaemia (CML). However, CML patients with methylated *SFRP1* correlated with imatinib therapy resistance as well as additional second Philadelphia chromosome abnormalities [93]. Moreover, expression of antagonists *SFRP1* and *WIF1* was shown to sensitize chronic myeloid leukemia (CML) cells to tyrosine kinase inhibitors [94]. In in vitro experiments involving K562 cells stably expressing *SFRP1*, the sensitivity toward imatinib, dasatinib, and nilotinib, were 75%, 43%, and 48% more sensitive, respectively, when compared to empty vector-transfected controls [94].

In another study, the dual role of *SFRP1* as a biomarker for chemoresistance was demonstrated. Upregulated *SFRP1* was observed in topotecan-resistant ovarian cancer patients, whereby its mRNA level was already high before topotecan treatment [95]. While the study did not investigate the treatment effects, the authors propose that *SFRP1* might act as an oncogene in patients treated with topotecan, or in a contrary manner, the cancer cells with high *SFRP1* expression could be inherently more resilient to the elimination of rapidly proliferating tumor cells [95]. Collectively, these findings suggested that *SFRP1* may offer therapeutic benefits and assist patient stratification in chemotherapy treatment. Evidently, more research is necessary to clarify the role of *SFRP1* in modulating response to cancer treatment.

### 4.2. SFRP1 and Disease Prognosis

*SFRP1* hypermethylation or downregulation are also associated with poor prognosis in several cancers. Kaplan–Meier analysis of nasopharyngeal cancer showed patients that exhibit low *SFRP1* had significantly worse overall survival (OS), disease-free survival (DFS), and distant-metastasis-free survival (DMFS) as compared to patients that have high expression of this gene [8]. Davaadorj and his team have discovered that the loss of *SFRP1* is associated with poor prognosis in hepatocellular carcinoma (HCC). They examined the expression status of *SFRP1* in 63 pairs of human HCC and later they found low *SFRP1* is associated with larger tumor size and vascular invasion as compared to the positive *SFRP1* patients [96]. Similarly, patients with low *SFRP1* expression had a poor OS in glioblastoma multiforme (GBM), relative to the positive *SFRP1* patients, which seem to have favorable prognosis [97]. *SFRP1* methylation is also associated with ovarian cancer recurrence and short overall survival [98].

A recent study by Kumar et al. showed promoter methylation of *SFRP1* is associated with lymph node metastasis and poor mean overall survival (OS) in CRC. The majority of CRC samples in this study are methylated at *SFRP1* with 72.2% methylation frequency [99]. On the contrary, Liu and colleagues postulated that the co-hypermethylation of *SFRP1* and *SFRP2* were suggested as independent prognostic predictors of survival advantage in postoperative CRC patients [100], whereby silencing of *SFRP1* and *SFRP2* by hypermethylation lead to a better prognosis. Meanwhile, in an in vivo study to correlate gut microbiota with epigenetic signature, CRC-associated microbiota induced higher numbers of hypermethylated genes in the murine colonic mucosa, among which *SFRP1* was also hypermethylated [101]. Additional confirmation was obtained in 1000 patients, further demonstrating that CRC-associated dysbiosis may promote CRC development via epigenome dysregulation. The authors suggested gene methylation as a marker for CRC to predict the efficacy of prebiotic supplementation in average-risk individuals [101]. Low *SFRP1* expression may also concurrently activate the Wnt pathway with *WIF1* gene [102]. High expression of *WIF1* was shown to be significantly correlated with big tumor diameters and deep invasion of tumor cells. However, the co-expression of high *SFRP1* and *WIF1* may also increase favorable OS and is associated with low TNM stage in CRC [102], postulating the role of *WIF1* as an oncogene, while *SFRP1* seemed to be an oncosuppressor, despite both being secreted Wnt antagonists. Nevertheless, the underlying mechanism is yet to be identified.

Hypermethylation of *SFRP1* occurs in 37.5% of acute myeloid leukemia (AML), whereby patients that exhibit intermediate-risk karyotyping with concurrent methylated *SFRP1* showed poor prognosis, especially in the subgroup 60 years old and younger patients [103]. Association between *SFRP1* hypermethylation and poor prognosis was further supported by another study conducted among non-M3 AML patients [51].

As mentioned earlier, *SFRP1* possess divergent roles depending on the context. This is shown by a study by Qu et al., which demonstrated that gastric patients with high *SFRP1* exhibit poor prognosis [12]. Moreover, this study also reveals high *SFRP1* was significantly associated with lymph node metastasis and worse five-year OS. Using gastric cancer cell line models, they further demonstrated that the *SFRP1* upregulation activates the TGF-β signaling pathway, hence inducing cell proliferation, EMT process, and cell invasion [12]. Additionally, upregulated *SFRP1* in metastatic renal cell carcinoma (RCC) was also involved in invasiveness of the metastatic cells. Knockdown of the endogenous *SFRP1* in this study has been showed to reduce the invasive properties of metastatic RCC [11]. On top of that, enriched *SFRP1* was also found in the metastatic osteosarcoma [104].

### 4.3. SFRP1 in Clinical Trials

Despite *SFRP1* being a well-known gene studied in human cancer, there are only a handful of clinical trials, which involve *SFRP1*. Based on the hypothesis that long-term exercise may cause changes in the serum levels of *SFRP1* in patients with breast cancer, a pilot trial (NCT02895178) was conducted [105]. Thirty breast cancer survivors from Wonju Severance Christian Hospital were enrolled and randomized to two different groups: 12 weeks exercise program and control group. Then, *SFRP1* expression in the serum was measured for the pre- and post-treatment in both groups. This pilot study is the first study to show a decrease in serum *SFRP1* levels in patients with breast cancer due to exercise training. The decrease of *SFRP1* is accompanied by improving body composition such as a decrease in body fat percentage and visceral fat areas. This demonstrates that decreased serum level of *SFRP1* improves the physical fitness of breast cancer survivor patients [105].

NCT01214681 is a clinical trial looking into the impact of the role of non-digestible carbohydrates (NDCs) in CRC chemoprevention [106]. A number of novel biomarkers of diet-related CRC risk measured in cancer tissue biopsies and in stool were developed, including *SFRP1*. The effects of supplementing healthy individuals with two NDCs, which are resistant starch (RS) and polydextrose, on fecal calprotectin concentrations and the expression of 12 Wnt-related genes including *SFRP1,* were investigated [107]. In addition, the trial also seeks to determine whether the effects on *SFRP1* expression are regulated via the epigenetic mechanisms, which include DNA methylation and microRNA expression. Although NDC supplementation did not influence fecal calprotectin concentration, *SFRP1* expression was significantly reduced by RS, which could result in increased Wnt pathway activity. However, RS and polydextrose neither affect *SFRP1* methylation nor alter the expression of 10 microRNAs that were predicted to target this gene. This suggests another unknown mechanism that reduces *SFRP1* expression. Nevertheless, the effects on Wnt pathway activity and downstream functional effects in the healthy colorectal mucosa warrants further investigation.

*SFRP1* is among the biomarkers studied in glioblastoma clinical trial NCT00822458 [108]. The phase I clinical trial is investigating the side effects and best dose of GDC-0449, a Hedgehog signaling antagonist (Vismodegib) in treating young patients with recurrent medulloblastoma or patients who did not respond to previous treatment. Blood samples are collected periodically for pharmacokinetic studies. Archived tumor tissues were collected and analyzed for the expression of genes that activate the Hedgehog (e.g., *Gli1*, *Gli2*, *SFRP1*, *ATOH1*, and *PTCH2*) or Wnt (e.g., *DKK2* and *DKK4*) cell signal pathways via in situ hybridization and real-time PCR. At the time this review was written, there is no published finding of this trial yet; therefore, the exact role of *SFRP1* is undetermined.

## 5. *SFRP1* in Patents

Despite the limited involvement of *SFRP1* in cancer clinical trials, research evidence has shown the potential of *SFRP1* as a diagnostic and pharmacogenetics marker, and this has led to several patents being applied or granted. For instance, Baylin and colleagues from John Hopkins University were granted a patent for methods of identifying epigenetically silenced genes that are associated with cancer in 2010 [109]. The method is demonstrated by the identification of 74 genes that are epigenetically silenced in CRC cells, including identification of methylation silencing of *SFRP1* and its families such as *SFRP2*, *SFRP4*, and *SFRP5*. This patent covers three categories, which are C12Q1/6886 nucleic acid products used in the analysis of nucleic acids, C12Q1/6809 methods for determination or identification of nucleic acids involving differential detection, and C12Q2600/154 methylation markers.

MDxHealth SA filed for a patent of a kit involving *SFRP1* in identifying and diagnosing cancer based on *SFRP1* methylation status [110]. In their claim, detection of the epigenetic change of *NDRG4* and at least one gene among 17 other genes including *SFRP1* indicate a predisposition to gastrointestinal cancer. Granted in 2016, the kit also described the pharmacogenetic methods for determining suitable treatment regimens for cancer. Combination of *NDRG4* and *SFRP1* is able to predict the response of cancer treatment with a DNA damaging agent, DNA methyltransferase inhibitor, and/or an HDAC inhibitor [110]. Patients with methylated *NDRG4* and *SFRP1* will respond better to these therapeutic agents. The epigenetic changes of *NDRG4* and *SFRP1* are also indicative of the histopathological stage of gastrointestinal cancer [110]. This patent includes six categories of claims, which are C12Q1/6886 nucleic acid products used in the analysis of nucleic acids; C12Q2600/106 pharmacogenomics; C12Q2600/112 disease subtyping, staging, or classification; C12Q2600/118 prognosis of disease development; C12Q2600/154 methylation markers; C12Q2600/158 expression markers; and C12Q2600/16 primer sets for multiplex assays.

More recently, Lothe and colleagues identified *CDO1*, *DCLK1*, and *SCAN18* as novel, frequently methylated genes in cholangiocarcinoma, in addition to the previously reported *SFRP1* gene [111]. The combination of *CDO1*, *DCLK1*, *ZSCAN18*, and *SFRP1* reach a sensitivity of 87% and specificity of 100% in fresh, frozen, and archival material. The group was granted a patent in 2017 under two categories, which are C12Q1/6886 nucleic acid products used in the analysis of nucleic acids and C12Q2600/154 methylation markers.

## 6. Conclusions

In this review, we have provided the latest evidence of epigenetic of *SFRP1* and its divergent roles in carcinogenesis, highlighted its pharmacogenomics properties, and deliberated on the clinical trials and patents involving *SFRP1*. SFRP1 functions as a negative regulator of Wnt signaling; therefore, it has an important role in carcinogenesis. Accumulating evidence suggests that epigenetic regulation contributes to the silencing of *SFRP1*. Indeed, in many cancers, *SFRP1* is downregulated via promoter hypermethylation. Restoration of *SFRP1* expression via demethylating drugs or over-expression experiments increases sensitivity toward various chemotherapeutic agents. Given the importance of *SFRP1* in cancer-related pathways, HDACi and DNMTi appear as promising epigenetic therapy to reverse *SFRP1* methylation. Interestingly, a combination of chemotherapeutic and epigenetic drug is also highly synergistic at inhibiting cancer cells than single agent alone.

On the other hand, owing to its dual roles, knockdown of *SFRP1* also renders certain cancer cells to be more resistant toward selected chemotherapies as well as radiotherapy. Nevertheless, the clinical utility of epigenetic drugs required extensive investigations and proper utilization, as these drugs are toxic and non-specific gene modulators. Although a lot of progress has been made regarding the regulation and function of *SFRP1* in the normal and cancer cells, many questions remain unanswered. Henceforth, the investigation into the contradictory roles of *SFRP1*, particularly in cancer prognosis and its pharmacogenomics utilities, are indispensable.

## Figures and Tables

**Figure 1 cancers-12-00445-f001:**
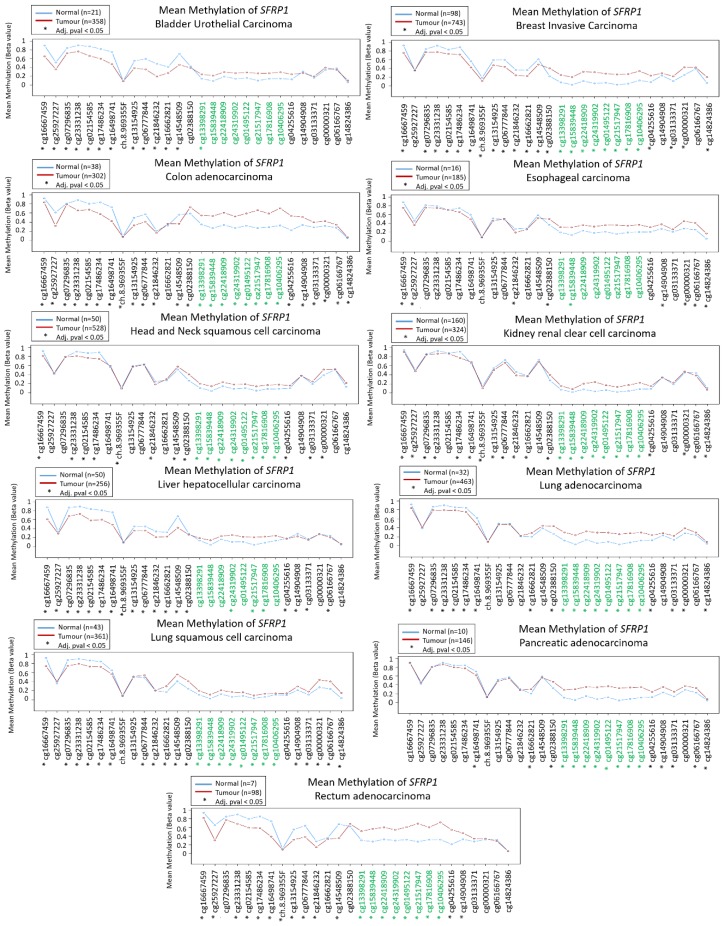
Methylation signatures of *SFRP1* in Cancer Genome Atlas (TCGA) datasets. Only selected studies with significant *SFRP1* differential methylation were shown. Green indicates CpG islands and the figure was generated using Wanderer [44].

**Figure 2 cancers-12-00445-f002:**
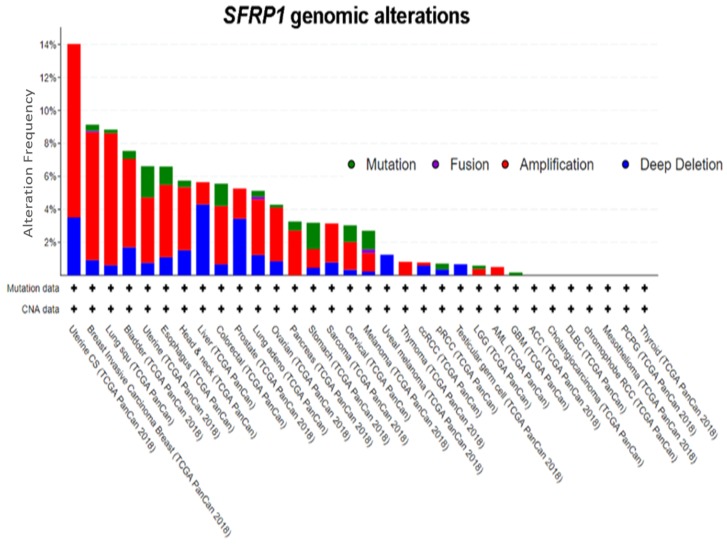
Genomic alterations in SFRP1. Amplification is the most frequent genomic event in *SFRP1*. Abbreviations: ACC, adrenocortical carcinoma; BLCA, bladder urothelial carcinoma; BRCA, breast invasive carcinoma; CESC, cervical squamous cell carcinoma and endocervical adenocarcinoma; CHOL, cholangiocarcinoma; COAD, colon adenocarcinoma; COADREAD, colorectal adenocarcinoma; DLBC, lymphoid neoplasm diffuse large B-cell lymphoma; ESCA, esophageal carcinoma; GBM, glioblastoma multiforme; GBMLGG, glioblastoma multiforme Brain lower grade glioma; HNSC, head and neck squamous cell carcinoma; KICH, kidney chromophobe; KIPAN, pan-kidney cohort (KICH+KIRC+KIRP); KIRC, kidney renal clear cell carcinoma; KIRP, kidney renal papillary cell carcinoma; AML, acute myeloid leukemia; LGG, brain lower grade glioma; LIHC, liver hepatocellular carcinoma; LUAD, lung adenocarcinoma; LUSC, lung squamous cell carcinoma; MESO, mesothelioma; OV, ovarian serous cystadenocarcinoma; PAAD, pancreatic adenocarcinoma; PCPG, pheochromocytoma and Paraganglioma; PRAD, prostate adenocarcinoma; READ, rectum adenocarcinoma; SARC, sarcoma; SKCM, skin cutaneous melanoma; STAD, stomach adenocarcinoma; STES, esophagus–stomach cancers; TGCT, testicular germ cell tumors; THCA, thyroid carcinoma; THYM, thymoma; UCEC, uterine corpus endometrial carcinoma; UCS, uterine carcinosarcoma; UVM, uveal melanoma [55,56].

**Figure 3 cancers-12-00445-f003:**
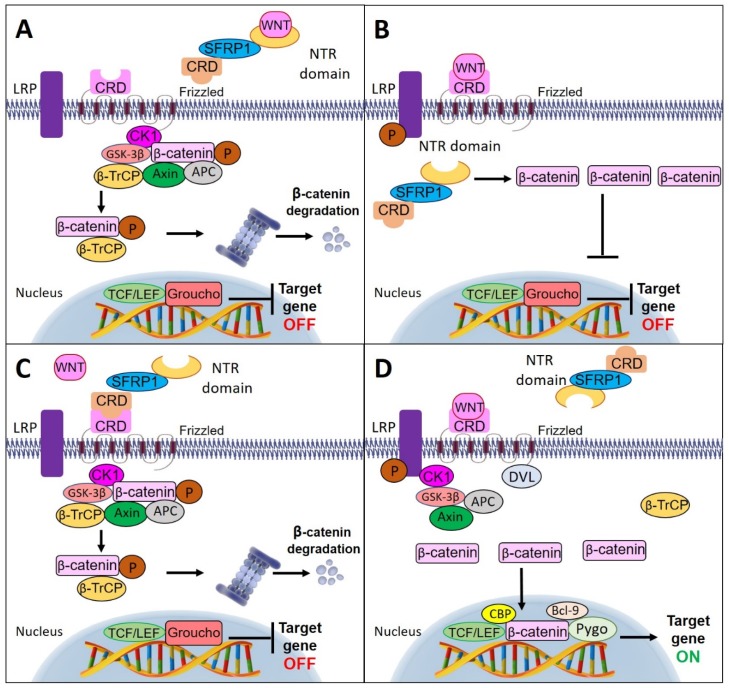
Schematic diagram of multiple mechanisms of SFRP1 involvement in Wnt signaling pathway. (**A**) SFRP1 inhibits Wnt activity by directly binding to the ligand of Wnt protein through its netrin (NTR) domain. (**B**) SFRP1 antagonizes the Wnt signaling pathway by bypassing the interaction with the Wnt ligand and directly binding to the cytoplasmic β-catenin. (**C**) SFRP1 binds directly with the Fz receptor through the cycteine-rich domain (CRD) motif preventing the binding of Wnt ligands to the receptor. This causes the β-catenin molecules to combine with the destruction complex and undergo phosphorylation by glycogen synthase kinase (GSK-3β) and casein kinase (CK1). The phosphorylated β-catenin triggers ubiquitination by β-TrCP and initiates proteasomal degradation. (**D**) In the absence of SFRP1, the Wnt ligand binds to the Fz receptor and results in the recruitment of Dvl and destruction complex to the membrane. The destruction complex is inactivated by Dvl polymers and leads to the accumulation of β-catenin molecules in the cytoplasm. Then, the accumulated β-catenin molecules enter the nucleus and recruit histone-modifying coactivators, Pygo and Bcl-9, to activate Wnt target genes’ transcription.

**Table 1 cancers-12-00445-t001:** Other microRNAs associated with secreted frizzled-related protein 1 (*SFRP1*) transcriptional silencing.

MicroRNA	Association with *SFRP1* Downregulation	Cancer	References
miR-1207	Upregulation of miR-1207 activated Wnt signaling pathway by inhibiting *SFRP1* activity and is inversely correlated with patients’ overall survival.	Ovarian	[29]
miR-454-3p	Over-expression of miR-454-3p reduces the *SFRP1* activity by targeting 3’UTR. High expression of miR-454-3p correlates with shorter relapse-free survival of breast cancer.	Breast	[30]
miR-196a-1	High expression of exosomal miR 196a-1 is associated with poor survival of gastric cancer. Ectopic miR 196a-1 expression promotes invasion of low invasive gastric cancer cells by binding to 3’UTR of *SFRP1*.	Gastric	[31]
miR-1301-3p	Significantly upregulated in prostate cancer and targets Wnt pathway inhibitors, *SFRP1* and *GSK3β*, by directly binding to the 3’UTR. High expression of miR-1301-3p suppresses the *SFRP1* expression and promotes the expansion of prostate cancer stem cells.	Prostate	[32]
miR-1260b	Potential candidate oncogenic miRNA in prostate cancer. Treatment with Genistein significantly downregulated miR-1260b and induces expression of *SFRP1* and *SMAD4* via DNA demethylation and histone modification.	Prostate	[33]
miR-582-3p	High expression correlates with the overall and recurrence-free survival of non-small cell lung carcinoma (NSCLC). Upregulated miR-582-3p inhibits *SFRP1*, *AXIN2*, and *DKK3* expression and leads to the interaction of β-catenin to TCF-4 and subsequently activates Wnt signaling pathway. Activated Wnt pathway is associated with tumor recurrence in NSCLC.	NSCLC	[34]

**Table 2 cancers-12-00445-t002:** Regulation of *SFRP1* expression by DNA methylation in cancer.

Cancer	Description	References
Breast	Methylation of *SFRP1* was significantly different according to the breast cancer molecular subtypes. Low methylation of *SFRP1* was detected in basal-like subtype compared to luminal A, luminal B, and HER2 subtypes. *SFRP1* may potentially serve as epigenetic biomarker.	[9]
Breast	Hypermethylation of *SFRP1* was frequently found in breast cancer and caused a reduction of *SFRP1* expression. Methylation of *SFRP1* indicates poor prognosis in ER+/HER2.	[45]
Breast	Aberrant methylation of *SFRP1* was observed in 96 breast cancer Chinese patients. This study also showed that methylation of *SFRP1* negatively regulates the expression level.	[46]
Glioma	Hypermethylation of *SFRP1* was associated with poor survival of patients (within 1–3 months after tumor resection). Low methylation of *SFRP1* was discovered in the longer survival group. Moreover, methylated *SFRP1* frequently occurred in the patients that exhibit higher-grade tumors. This study suggests the hypermethylated *SFRP1* as a potential prognostic biomarker in glioma.	[47]
Glioma	The methylation level of *SFRP1* increases with higher astrocytoma grades and is the highest in glioblastoma. *SFRP1* is epigenetically silenced and involved in the progression of glioma.	[48]
Glioma	Hydrogen peroxide reverses the methylation of *SFRP1* in U251 glioma cells. The demethylation of *SFRP1* leads to the activation of this gene and was partially involved in the apoptosis process of the hydrogen-peroxide-induced U251 cells.	[49]
Ovarian	Loss of SFRP1 protein expression caused by promoter hypermethylation was observed in the subset of high-grade serous ovarian carcinoma.	[20]
Ovarian	Aberrant methylation and low expression of *SFRP1* were associated with epithelial ovarian cancer. The activation of this gene inhibits the tumor growth through inactivation of the Wnt signaling pathway. This study further showed that the *SFRP1* over-expression in the in vivo model could inhibit the growth of cancer cells.	[50]
Acute myeloid leukemia (AML)	Aberrant methylation of *SFRP1* was observed in 30.2% of non-M3 AML patients, and *SFRP1* expression was negatively correlated with its promoter methylation.	[51]
Malignant pleural mesothelioma	Long-term asbestos exposure led to hypermethylation of *SFRP1* and reduced gene expression in the mesothelium.	[52]

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
