# Peer review of "Epigenetics of SFRP1: The Dual Roles in Human Cancers"

_cancers, 2020, doi:10.3390/cancers12020445_

Round 1

Reviewer 1 Report

Authors describe the latest evidence of the divergent roles of SFRP1 in human cancer. This manuscript is well organized and timely appropriate in this field. And then I would like to suggest few minor revision.

Minor:

Please make a full sentence: line 276-278 in page 8. I am not able to read some characters due to small font in Fig 3. Place Table 2 in one page. Correct the exression to the expression: line 386 in page 12.  

Author Response

Response to Reviewer 1 Comments

Point 1: Please make a full sentence: line 276-278 in page 8.

Response 1: We have completed the sentence. Please refer to line number 137-138.

Point 2: I am not able to read some characters due to small font in Fig 3.

Response 2: We have revised the figure accordingly. To accommodate comments from the Editor, we have renamed figure 3 as figure 1 on page 5.

Point 3: Place Table 2 in one page.

Response 3: Table 2 was edited accordingly. Please refer to line 147 on page 6.

Point 4: Correct the exression to the expression: line 386 in page 12.  

Response 4: We have corrected it accordingly. Please refer to line 401 on page 14.

We sincerely hope that the corrections we made strengthen the manuscript. Thank you for your valuable time with our manuscript and we thank your effort with sincerity.

Reviewer 2 Report

I suggest to modify the title of the review in:

SFRP1: a dual role in human cancers.

In addition, the manuscript needs proofreading to correct some editing errors.

Author Response

Response to Reviewer 2 Comments

Point 1: I suggest to modify the title of the review in: SFRP1: a dual role in human cancers.

Response 1: Thank you for the suggestion, we have modified the title accordingly. Please refer to line number 2.

Point 2: In addition, the manuscript needs proofreading to correct some editing errors.

Response 2: We have carefully proofread the manuscript and corrected the editing errors.

We sincerely hope that the corrections we made strengthen the manuscript. Thank you for your valuable time with our manuscript and we thank your effort with sincerity.

Reviewer 3 Report

The manuscript titled "SFRP1: The divergent roles in human cancers" by Baharudin R. et al. seeks to make an updated review (last review in 2014) that resume divergent roles of secreted frizzled-related protein 1 (SFRP1) gene in cancer and the association between SFRP1 and chemoresistance.

The review is well organized in different thematic sections and the bibliography is carefully updated, but it is written in bad English and is full of grammatical mistakes in many parts (especially in sections 1-2-3) that make it difficult to understand the contents. Furthermore, the authors must pay more attention to the correct use of italics to indicate genes instead of proteins, because in many parts they want to refer to proteins but the protein name is written in italics (i.e. lines 71-72).

In conclusion, the manuscript could be interesting for the scientific community working on SFRP protein family, but the current version needs a major revision to make the content fully comprehensible.

Author Response

Response to Reviewer 3 Comments

Dear Reviewer,

Please find the responses to your comments and suggestions below. Kindly view the revised manuscript in using Track Changes – Simple Markup.

Point 1: The manuscript titled "SFRP1: The divergent roles in human cancers" by Baharudin R. et al. seeks to make an updated review (last review in 2014) that resume divergent roles of secreted frizzled-related protein 1 (SFRP1) gene in cancer and the association between SFRP1 and chemoresistance.

The review is well organized in different thematic sections and the bibliography is carefully updated, but it is written in bad English and is full of grammatical mistakes in many parts (especially in sections 1-2-3) that make it difficult to understand the contents.

Response 1: We have edited the contents accordingly throughout the manuscript.

Point 2: Furthermore, the authors must pay more attention to the correct use of italics to indicate genes instead of proteins, because in many parts they want to refer to proteins but the protein name is written in italics (i.e. lines 71-72).

Response 2: We have edited the contents accordingly throughout the manuscript. Please refer to line 196-197, 200-202, 204, 216, 223, 245, 259, 261-262, 264-267, 269-270, 275, 281-282, 284-285, 288, 295 and 297-298 on page 9-12.

Point 3: In conclusion, the manuscript could be interesting for the scientific community working on SFRP protein family, but the current version needs a major revision to make the content fully comprehensible.

Response 3: Thank you for your comment and we have revised accordingly.

We sincerely hope that the corrections we made strengthen the manuscript. Thank you for your valuable time with our manuscript and we thank your effort with sincerity.

Round 2

Reviewer 3 Report

Baharudin et al. made an appropriate reorganization and an accurate english language correction of the previous version of the review and now the revised version is very well organized and the contents are properly described.

This manuscript is now an interesting and updated review in which the authors discuss in detail the dual roles of SFRP1 in cancers, as tumor suppresor or promoter.

I have only a minor concern of a primarily editorial nature:

it is not exactly correct to indicate that a gene, in this case SFRP1, belongs to a protein family (Lines 13-14 and 33-34).